# GNNCert: Deterministic Certification of Graph Neural Networks against Adversarial Perturbations

Zaishuo Xia[1,*], Han Yang[2,*], Binghui Wang[3,†], Jinyuan Jia[4,†]
[1]Renmin University of China, [2]Sichuan University,
[3]Illinois Institute of Technology, [4]The Pennsylvania State University

## Abstract

*Graph classification*, which aims to predict a label for a graph, has many real-world applications such as malware detection, fraud detection, and healthcare. However, many studies show an attacker could carefully perturb the structure and/or node features in a graph such that a graph classifier misclassifies the perturbed graph. Such vulnerability impedes the deployment of graph classification in security/safety-critical applications. Existing empirical defenses lack formal robustness guarantees and could be broken by adaptive or unknown attacks. Existing provable defenses have the following limitations: 1) they achieve sub-optimal robustness guarantees for graph structure perturbation, 2) they cannot provide robustness guarantees for arbitrarily node feature perturbations, 3) their robustness guarantees are probabilistic, meaning they could be incorrect with a non-zero probability, and 4) they incur large computation costs. We aim to address those limitations in this work. We propose GNNCert, a certified defense against both graph structure and node feature perturbations for graph classification. Our GNNCert provably predicts the same label for a graph when the number of perturbed edges and the number of nodes with perturbed features are bounded. Our results on 8 benchmark datasets show GNNCert outperforms three state-of-the-art methods[1].

## 1 Introduction

In graph classification, a graph classifier takes a graph as input and predicts a label for it. Graph classification is a fundamental task for graph analytics, which has been widely used in many real-world applications such as fraud detection (Weber et al., 2019), malware detection (Kong & Yan, 2013; Hassen & Chan, 2017; Yan et al., 2019), and healthcare (Li et al., 2017; Chen et al., 2018). Graph neural networks (Kipf & Welling, 2016) are widely used as graph classifiers as they achieve state-of-the-art classification accuracy for graph classification. Despite their superior performance, many studies (Dai et al., 2018; Tang et al., 2020; Chen et al., 2021; Mu et al., 2021b; Zhang et al., 2021a; Wan et al., 2021b; Zhang et al., 2022; Mu et al., 2021a; Wang et al., 2022; 2023) show graph classification is vulnerable to adversarial perturbations, where an attacker could arbitrarily perturb the graph structure and/or node features of a testing graph such that a graph classifier predicts an incorrect label for the testing graph. Such attacks significantly impede the deployment of the graph classification for security- and safety-critical applications such as fraud detection.

Many defenses were proposed to defend against adversarial perturbations to graph classification. In particular, those defenses can be classified into *empirical defenses* (Chen et al., 2020; Zhang & Lu, 2020; Zhao et al., 2021) and *certified defenses* (Bojchevski et al., 2020; Wang et al., 2021; Zhang et al., 2021b). Empirical defenses cannot provide formal robustness guarantees, i.e., they cannot guarantee the graph classification performance under arbitrary attacks, and thus lead to a cat-and-mouse game between the attacker and defender. As a result, they could be broken by advanced attacks, as shown in Mujkanovic et al. (2022). By contrast, certified defenses could guarantee the graph classification performance under arbitrary attacks, once the number of perturbed edges and number

---

[1]*Equal contribution, †Corresponding authors. Zaishuo and Han performed this research when they were interns. The code is available at `https://github.com/XiaFire/GNNCERT`.

of nodes with perturbed features are bounded. However, existing certified defenses (Bojchevski et al., 2020; Wang et al., 2021; Zhang et al., 2021b) suffer from the following limitations. First, they cannot provide robustness guarantees when an attacker can arbitrarily perturb the features of nodes. Second, they achieve sub-optimal robustness guarantees for graph structure perturbation as shown in our experimental results. Third, their robustness guarantees are probabilistic, i.e., their robustness guarantees could be incorrect with a certain probability. Fourth, their computation costs are large. We aim to address all these limitations in this work.

**Our contribution:** In this work, we propose GNNCert, the first certified defense against *both graph structure and node feature perturbations* for graph classification with *deterministic* robustness guarantees. Given a testing graph and a graph classifier (called *base graph classifier*), we first use a hash function to divide the testing graph into multiple sub-graphs, then use the graph classifier to predict a label for each sub-graph, and finally use the majority vote to predict a label for the testing graph. We show the majority vote result is provably unaffected, i.e., GNNCert provably predicts the same label for a testing graph under arbitrary adversarial structure and feature perturbations, once the number of perturbed edges and number of nodes with perturbed features are bounded. Following Cohen et al. (2019), to improve the provable robustness guarantees of GNNCert, we also utilize sub-graphs created from training graphs to train the base graph classifier. Our experimental results show such a training method significantly improves the provable robustness guarantees.

We conduct comprehensive evaluations on 8 benchmark datasets (e.g., MUTAG (Debnath et al., 1991), NCI1 (Wale et al., 2008), PROTEINS (Borgwardt et al., 2005), COLLAB (Yanardag & Vishwanathan, 2015)) for graph classification. We use *certified accuracy* as the evaluation metric, which is a lower bound of the classification accuracy of a defense under arbitrary adversarial attacks with a bounded number of perturbed edges and/or nodes with perturbed features. We compare our GNNCert with state-of-the-art certification methods (Bojchevski et al., 2020; Wang et al., 2021; Zhang et al., 2021b), and results show GNNCert significantly outperforms them. Our major contributions are as follows:

- We propose GNNCert, a certified defense against both graph structure and node feature perturbations to graph classification.
- We derive the deterministic robustness guarantee of GNNCert.
- We extensively evaluate GNNCert on 8 benchmark datasets.

## 2 BACKGROUND AND RELATED WORK

**Adversarial attacks to graph classification:** Many existing studies (Dai et al., 2018; Tang et al., 2020; Chen et al., 2021; Mu et al., 2021b; Zhang et al., 2021a; Wan et al., 2021b; Zhang et al., 2022; Mu et al., 2021a; Wang et al., 2022; 2023) show graph classification is vulnerable to adversarial attacks. Given a testing graph, an attacker could perturb the graph structure and/or node features such that a graph classifier makes incorrect predictions for the perturbed testing graph. For instance, Wang et al. (2022) proposed to leverage bandits to design black-box attacks on graph classifications with guaranteed attack performance.

**Existing empirically robust defenses:** To defend against adversarial attacks, many empirical defenses (Chen et al., 2020; Zhang & Lu, 2020; Zhao et al., 2021) were proposed. The key limitation of those empirical defenses is that they cannot guarantee their performance under arbitrary adversarial perturbations, leading to a cat-and-mouse game between the attacker and defender. In other words, those empirical defenses could be broken by adaptive attacks (Mujkanovic et al., 2022).

**Existing provably robust defenses:** Existing certified defenses (Bojchevski et al., 2020; Jin et al., 2020; Wang et al., 2021; Zhang et al., 2021b) could show they provably predict the same label for a testing graph when the number of perturbed edges to the graph is bounded. For instance, Bojchevski et al. (2020) and Wang et al. (2021) generalized randomized smoothing (Lecuyer et al., 2019; Cohen et al., 2019) (a state-of-the-art method to certify robustness against adversarial examples) from the image domain to the graph domain. Zhang et al. (2021b) extended randomized ablation (Levine & Feizi, 2020c) to build provably robust graph classification against graph structure perturbations. Existing certified defenses for graph classification suffer from the following limitations. First, they can only provide robustness guarantees for graph structure perturbations. However, in practice, an

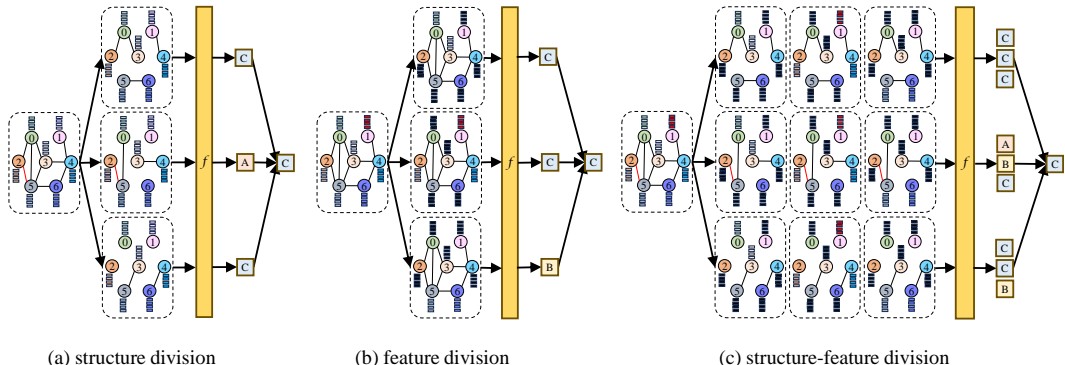

(a) structure division            (b) feature division            (c) structure-feature division

**Figure 1: Overview of GNNCert. The black node features mean special values, e.g., zero.**

attacker may perturb node features to perform attacks. Second, they achieve sub-optimal certified robustness guarantees for graph structure perturbations as shown in our experimental results. Third, they incur large computation costs. Fourth, their robustness guarantees are probabilistic, which means their guarantees could be incorrect with a certain probability. We aim to address those limitations. We note that the defense proposed by Jin et al. (2020) can only be applied to specific architectures.

## 3  OUR GNNCERT

### 3.1  GRAPH CLASSIFICATION SETUP

Given a graph $G = (V, E, X)$, where $V = \{v_1, v_2, \cdots, v_n\}$ is a set of $n$ nodes, $E$ is a set of edges connecting nodes in $V$, and $X_v$ denote the feature vector of the node $v \in V$. Given a training dataset $\mathcal{D}_{tr}$ that contains a set of training graphs and their ground truth labels, we can use it to train a graph classifier $f$. Given a testing graph $G$, we could use the graph classifier $f$ to predict a label for it. For simplicity, we use $f(G)$ to denote the predicted label for the graph $G$.

### 3.2  THREAT MODEL

**Attacker's goal, background knowledge, and capability:**  Given a testing graph $G$ and a graph classifier $f$, an attacker's goal is to perturb the graph structure and/or node features of $G$ such that the graph classifier $f$ makes an incorrect prediction for the perturbed testing graph. As we focus on provable defenses, we consider the strongest attack. That is, the attacker knows all information (e.g., the edges, nodes, and node features) about $G$, and all information (e.g., the model parameters, architecture) about $f$. We further consider the attacker can manipulate both the graph structure and node feature. For graph structure perturbations, the attacker could arbitrarily add or delete a certain number of edges in the testing graph $G$. For node feature perturbations, the attacker could arbitrarily perturb the features of a certain number of nodes in $G$.

**Defender's goal:**  The defender aims to build a provably robust graph classifier such that it provably predicts the same label for the testing graph $G$ when the number of perturbed edges and/or number of nodes with perturbed features are no larger than a threshold (called *certified perturbation size*).

### 3.3  OUR GNNCERT

GNNCert consists of three main steps: 1) divide a graph into different sub-graphs; 2) build an ensemble classifier on the sub-graphs; and 3) derive the formal robustness guarantees of the ensemble classifier against graph structure perturbations, node feature perturbations, as well as both graph structure and node feature perturbations. Figure 1 shows an overview of GNNCert.

#### 3.3.1  DIVIDING A GRAPH INTO SUB-GRAPHS

Our idea is to use a hash function $\mathcal{H}$ ($\mathcal{H}$ does not depend on the edge structure/node features) to divide $G$ into different sub-graphs. A hash function (e.g., MD5) takes a string as input and produces an integer. We leverage the node IDs to divide the testing graph $G$ into multiple sub-graphs. Let

$\text{ID}_v$ be the node ID of a node $v$. We can transform $\text{ID}_v$ into a string, denoted as $S_v$. For instance, $S_v$ could be "3" when the node ID of $v$ is 3. Given $S_v$'s for every node $v \in V$, we propose three ways, namely *structure division*, *feature division*, and *structure-feature division*, to divide the testing graph $G$ into different sub-graphs. In particular, the structure division, feature division, and structure-feature division, enable our GNNCert to defend against graph structure perturbations, node feature perturbations, as well as both graph structure and node feature perturbations, respectively. Next, we will discuss the three division methods in detail.

**Structure division:** Suppose an attacker aims to perturb the graph structure by adding/deleting edges in $G$. To defend against this attack, we can divide edges in $E$ into $T_s$ groups using the hash function $\mathcal{H}$, where $T_s$ is a hyper-parameter. First, we can compute a group index for every edge $(v_i, v_j) \in E$ based on the concatenation of $S_{v_i}$ and $S_{v_j}$. For simplicity, we denote $\mathcal{H}(S_{v_i} \oplus S_{v_j})\%T_s + 1$ as the group index for the edge $(v_i, v_j)$, where $\%$ is the modulo operation and $\oplus$ represents the concatenation of two strings. Then, we use $E^t$ to denote the set of edges whose group index is $t$, where $t = 1, 2, \cdots, T_s$, i.e., $E^t = \{(v_i, v_j) \in E | \mathcal{H}(S_{v_i} \oplus S_{v_j})\%T_s + 1 = t\}$. Given each $E^t$, we can construct the each sub-graph $\mathcal{G}_t$ as $\mathcal{G}_t = (V, E^t, X)$. We call this method *structure division* as we keep all the nodes and their features in each sub-graph while dividing edges into different sub-graphs.

**Feature division:** Suppose an attacker aims to perturb features of nodes in $V$. For this type of attack, we can divide features of nodes into $T_f$ groups using the hash function $\mathcal{H}$, where $T_f$ is a hyper-parameter. Likewise, we could first compute a group index $\mathcal{H}(S_v)\%T_f + 1$ for every node $v$ using the hash function $\mathcal{H}$. Then we use $X^t$ to denote the features of nodes whose group index is $t$, where $t = 1, 2, \cdots, T_f$. Then, we construct the sub-graph $\mathcal{G}_t$ as $\mathcal{G}_t = (V, E, X^t)$. Note that there exist many nodes whose feature vectors are not in $X^t$, and we set their feature vectors to be a special value, e.g., zero. We call this method *feature division* as we keep the entire graph structure in each sub-graph but divide node features into different sub-graphs.

**Structure-feature division:** In structure-feature division, we combine the above strategies to construct sub-graphs. In particular, we first use structure division to divide edges $E$ into $T_s$ groups $E^1, E^2, \cdots, E^{T_s}$ and use feature division to divide features $X$ into $T_f$ groups $X^1, X^2, \cdots, X^{T_f}$. Then we can construct $T_s \cdot T_f$ sub-graphs. In particular, we have $\mathcal{G}_t = (V, E^u, X^v)$, where $t = 1, 2, \cdots, T_s \cdot T_f$, $u = \lceil t/T_f \rceil$, and $v = t - (u - 1) \cdot T_f$. We call this method *structure-feature division* as we divide both graph structure and node features into different sub-graphs.

### 3.3.2 BUILDING AN ENSEMBLE GRAPH CLASSIFIER

Given a testing graph $G$, we can use structure (or feature or structure-feature) division to divide it into $N$ sub-graphs, where $N = T_s$ (or $N = T_f$ or $N = T_s \cdot T_f$). Given the $N$ sub-graphs $\{\mathcal{G}_t\}$ and a base graph classifier $f$, we can use $f$ to predict a label for each sub-graph. Note that we remove the node in the sub-graph if it simultaneously satisfies the following conditions: 1) it does not have any edge with all other nodes in the sub-graph, and 2) its feature vector is set to be a special value. The reason is that this kind of node does not provide any information for graph classification since it is not connected with any other nodes and its node feature is a special value. Suppose the set of all the possible classes is $\{1, 2, \cdots, C\}$. We use $N_c$ to denote the number of sub-graphs that are predicted as the class $c$ by the base graph classifier $f$, i.e., $N_c = \sum_{t=1}^{N} \mathbb{I}(f(\mathcal{G}_t) = c)$, where $c = 1, 2, \cdots, C$ and $\mathbb{I}$ is the indicator function. Then, we define our ensemble graph classifier $g$ as follows:

$$g(G) = \underset{c \in \{1, 2, \cdots, C\}}{\arg\max} N_c, \tag{1}$$

where a label with a smaller index is taken by our ensemble classifier when there are ties. We denote $l = g(G)$ as the predicted label. Then, $g$ can provably predict the same label for $G$ when the number of perturbed edges and/or number of nodes with perturbed features are bounded, as shown below.

### 3.3.3 DERIVING THE PROVABLE ROBUSTNESS GUARANTEES

Suppose we have an adversarially perturbed graph $G^p$. Similarly, we use $\mathcal{G}_1^p, \mathcal{G}_2^p, \cdots, \mathcal{G}_N^p$ to denote the $N$ sub-graphs created from $G^p$. Moreover, we denote by $N_c^p = \sum_{t=1}^{N} \mathbb{I}(f(\mathcal{G}_t^p) = c), \forall c \in \{1, 2, \cdots, C\}$, i.e., $N_c^p$ measures the number of sub-graphs created from $G^p$ that are predicted as the class $c$ by the base graph classifier $f$. Due to the perturbed node features or graph edges, some sub-graphs would be corrupted. For simplicity, we use $M$ to denote the total number of sub-graphs

in $\mathcal{G}_1^p, \mathcal{G}_2^p, \cdots, \mathcal{G}_N^p$ that are corrupted by the perturbed node features or graph edges. Then, we can derive the following lower or upper bounds:

$$N_c - M \leq N_c^p \leq N_c + M, \ \forall c \in \{1, 2, \cdots, C\}. \tag{2}$$

Our ensemble graph classifier $g$ still predicts the label $l$ for the perturbed graph $G^p$ if $N_l^p > \max_{c \in \{1,2,\cdots,C\} \setminus \{l\}} (N_c^p - \mathbb{I}(l < c))$, where the term $\mathbb{I}(l < c)$ stems from our tie breaking mechanism (i.e., we take a label with a smaller index where there are ties). Then we have the following theorem:

**Theorem 1.** *Given an arbitrary base graph classifier $f$, our ensemble graph classifier $g$ is as defined in Equation 1. Given a testing graph $G$, we can use our structure (or feature or structure-feature) division to divide it into $N$ sub-graphs. Suppose $N_c$ is the number of sub-graphs predicted as the label $c$ by the given base graph classifier $f$, where $c = 1, 2, \cdots, C$. Moreover, we assume $M$ is the total number of corrupted sub-graphs created from a perturbed graph $G^p$. Then, we have $g(G) = g(G^p) = l$ when the following condition is satisfied:*

$$M \leq M^p = \lfloor \frac{N_l - \max_{c \in \{1,2,\cdots,C\} \setminus \{l\}} (N_c - \mathbb{I}(l < c))}{2} \rfloor. \tag{3}$$

*Proof.* See Appendix A. □

We have the following remarks from our theorem:

- With structure division, each added or deleted edge results in one corrupted sub-graph. Thus, our GNNCert with structure division could tolerate up to $M^p$ added or deleted edges.

- With feature division, a single sub-graph is corrupted when an attacker *arbitrarily* perturb the features of a node. Thus, our GNNCert with feature division could tolerate up to $M^p$ nodes with adversarially perturbed features.

- With structure-feature division, $T_f$ (or $T_s$) sub-graphs are corrupted when an attacker adds/deletes an *arbitrary* edge (or *arbitrarily* perturb the features of a node). Thus, our GNNCert with feature division could tolerate up to $\lfloor M^p / T_f \rfloor$ perturbed edges (or $\lfloor M^p / T_s \rfloor$ of nodes with adversarially perturbed features).

- Our provable robustness guarantees hold for arbitrary adversarial attacks that perturb a bounded number of edges and/or node features. Moreover, our robustness guarantee is deterministic, i.e., the robustness guarantee is true with a probability of 1.

**Technical contributions:** To the best of our knowledge, we are the first to utilize a hash function to divide a graph into sub-graphs to build a certifiably robust graph classifier. Our GNNCert effectively addresses the aforementioned four limitations of state-of-the-art provable defenses for graph classification (Bojchevski et al., 2020; Wang et al., 2021; Zhang et al., 2021b). Particularly, it could resist structure perturbations, feature perturbations, as well as both structure and feature perturbations. Our GNNCert is effective and easy to implement, and thus could be widely applied in graph classification tasks that require provable robustness guarantees against adversarial attacks.

Our GNNCert is based on a general randomized smoothing framework (i.e., splitting & prediction & majority vote) (Lecuyer et al., 2019; Cohen et al., 2019). The key insight of the defense is based on 1) only a bounded fraction of predictions are corrupted when the perturbation is bounded, and 2) majority vote is intrinsically robust against corrupted voters (each prediction can be viewed as one voter). This framework was also utilized in previous certified defenses for adversarial patch attacks (Levine & Feizi, 2020b; Xiang et al., 2021), data poisoning attacks (Jia et al., 2021; Levine & Feizi, 2020a; Jia et al., 2022), $\ell_0$-norm adversarial examples for image and tabular data (Hammoudeh & Lowd, 2023), and many others (Zhang et al., 2023; Zeng et al., 2023; Pei et al., 2023). Our robustness guarantee derivation is also based on the intrinsic robustness of the majority vote, which follows previous certified defenses (Levine & Feizi, 2020b;a; Jia et al., 2022; Hammoudeh & Lowd, 2023). The key difference between different methods is that they create different voters (i.e., the voters have different meanings) for the majority vote. Our key contribution is to design three graph division methods (e.g., structure, feature, structure-feature division) tailored to the graph domain.

Our following theorem shows the tightness of our derived bound:

**Theorem 2.** *Without leveraging any information on the base graph classifier, our derived bound is tight, i.e., it is impossible to derive a tighter bound than ours.*

*Proof.* See Appendix B. □

# 4 EVALUATION

## 4.1 EXPERIMENTAL SETUP

**Datasets:** We use 8 benchmark datasets for graph classification in our evaluations: DBLP (Pan et al., 2013), DD (Dobson & Doig, 2003), ENZYMES (Hu et al., 2020), MUTAG (Debnath et al., 1991), NCI1 (Wale et al., 2008), PROTEINS (Borgwardt et al., 2005), REDDIT-B (Yanardag & Vishwanathan, 2015), COLLAB (Yanardag & Vishwanathan, 2015). Table 2 in the Appendix shows the statistics of those datasets. For each dataset, we randomly sample two-thirds of the graphs as the training dataset to train a base graph classifier and use the remaining graphs as the testing dataset.

**Base graph classifier:** We use state-of-the-art graph neural networks as the base graph classifier. Unless otherwise mentioned, we consider GIN (Xu et al., 2019) and utilize its publicly available implementation[2] in our experiments. To train a graph classifier, we use the Adam optimizer with a learning rate of 0.001 and a batch size of 32 for 1,000 epochs.

**Compared methods:** We compare our GNNCert with three state-of-the-art certification methods for graph classification: Zhang et al. (2021b), Wang et al. (2021), and Bojchevski et al. (2020). In particular, Zhang et al. (2021b) extends randomized ablation (Levine & Feizi, 2020c) from the image domain to the graph domain. Given a testing graph, Zhang et al. (2021b) create $S_z$ subsampled graphs, where each subsampled graph is obtained by sampling $\tau$ fraction of edges from a testing graph uniformly at random without replacement. Then, Zhang et al. (2021b) uses a base graph classifier to predict a label for each subsampled graph and take a majority vote over those predicted labels. Following Zhang et al. (2021b), we set $\tau = 10\%$ and set $S_z = 1,000$, and use subsampled training graphs to train the base graph classifier in our comparison.

Given a testing graph and a base graph classifier, Wang et al. (2021) and Bojchevski et al. (2020) first randomly perturb the graph structure of a testing graph to create $S_w$ and $S_b$ noisy graphs, then use the base classifier to predict a label for each noisy graph, and finally take a majority vote over the predicted labels. The key difference between Wang et al. (2021) and Bojchevski et al. (2020) is that they use different ways to perturb the graph structure vector to obtain a noisy graph. In particular, Wang et al. (2021) proposed to flip the connection status of two nodes with a probability $\beta$, e.g., if there is an edge between two nodes, the edge is deleted with a probability $\beta$. We set $\beta = 0.3$ by following Wang et al. (2021). By contrast, Bojchevski et al. (2020) utilizes two probabilities (denoted as $p_+$ and $p_-$) to create a noisy graph from a given testing graph. In particular, if there is an edge (or no edge) between two nodes, the edge is deleted (or added) with a probability of $p_-$ (or $p_+$). We set $p_- = 0.3$ (or $p_+ = 0.3$) in our experiments to be consistent with Wang et al. (2021) (i.e., on average, the number of perturbed edges is the same for those two methods). Following Wang et al. (2021) and Bojchevski et al. (2020), we set $S_w = 1,000$ and $S_b = 1,000$, i.e., we randomly create 1,000 noisy graphs for each testing graph for those two methods.

All the three compared methods utilize Monte-Carlo sampling to compute the certified perturbation size. As a result, the certified perturbation size is correct with a probability $1 - \alpha$, where $0 < \alpha < 1$. In our comparison, we set $\alpha = 0.001$ following those three methods. Note that we adopt the same graph neural network as the base graph classifier with our GNNCert in our comparison.

**Evaluation metric:** Following previous studies (Zhang et al., 2021b; Wang et al., 2021; Bojchevski et al., 2020), we use the certified accuracy as the evaluation metric. Given a perturbation size $S$ and a testing dataset $\mathcal{D}_{test}$, certified accuracy is defined as the fraction of testing inputs that 1) are correctly predicted, and 2) have a certified perturbation size that no smaller than $M$.

**Parameter setting:** Our GNNCert has the following hyperparameters: the hash function $\mathcal{H}$, the number of groups $T_s$ for structure division, and the number of groups $T_f$ for feature division. Unless otherwise mentioned, we use MD5 as the hash function and we set $T_s = 30$ and $T_f = 30$. We also conduct extensive ablation studies to show the impact of these hyper-parameters.

---

[2]https://github.com/weihua916/powerful-gnns

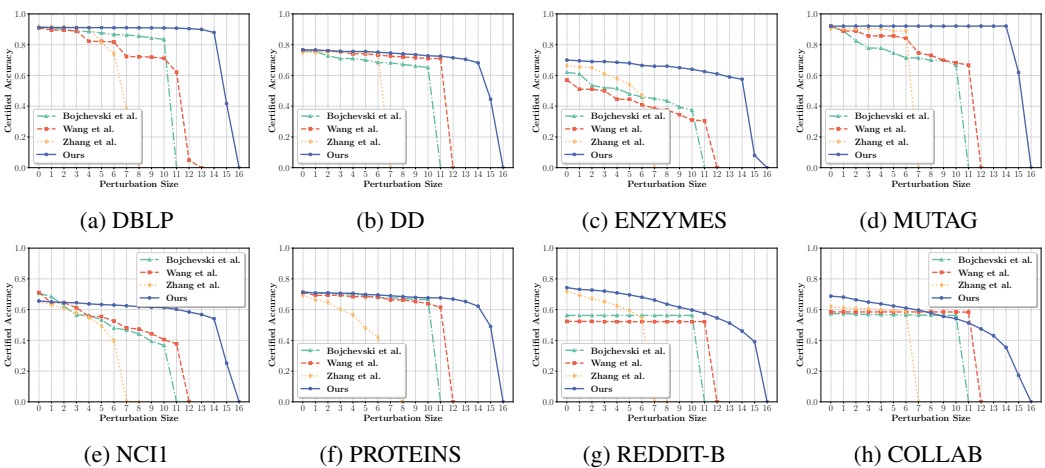

**Figure 2:** Comparing the certified accuracy of our GNNCert with existing certified defenses.

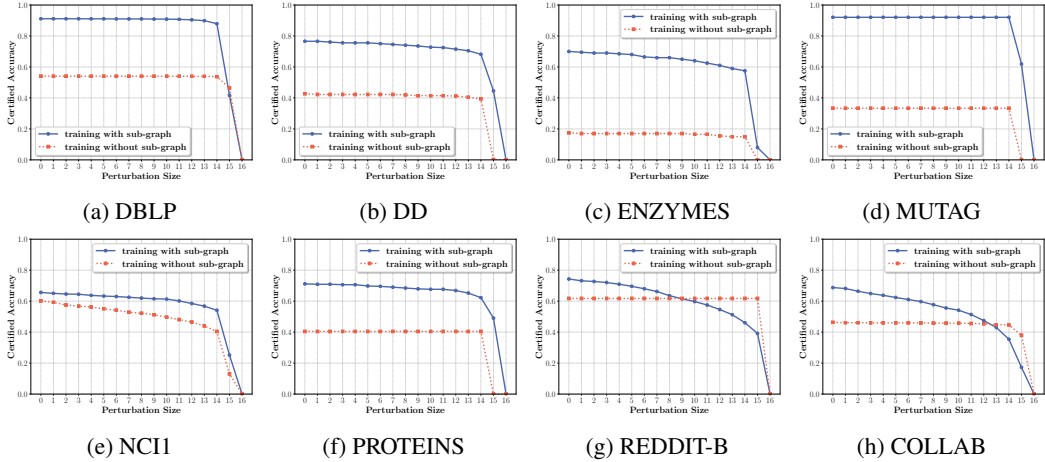

**Figure 3:** Comparing the certified accuracy of our GNNCert when the base graph classifier is trained with/without sub-graphs.

**Table 1:** Comparing the computation costs of GNNCert and existing defenses on MUTAG.

| Compared method | Training cost (s) | Testing cost (s) |
|---|---|---|
| Bojchevski et al. (2020) | 1,455 | 38 |
| Wang et al. (2021) | 1,344 | 48 |
| Zhang et al. (2021b) | 1,136 | 25 |
| GNNCert | 773 | 8 |

### 4.2 EXPERIMENTAL RESULTS

**Our GNNCert achieves better provable robustness guarantees than existing defense for graph structure perturbation:** Existing state-of-the-art certification methods (Zhang et al., 2021b; Wang et al., 2021; Bojchevski et al., 2020) for graph classification only provide robustness guarantees for graph structure perturbations. Thus, we only show the comparison results of our GNNCert with these methods for attacks that perturb the graph structure. Figure 2 shows the comparison of the certified accuracy of our GNNCert with existing defenses. We find that our GNNCert outperforms existing defenses in most cases. Our GNNCert is better than existing defenses because a perturbed edge could influence at most one sub-graph for our GNNCert but could influence multiple noisy graphs for existing defenses.

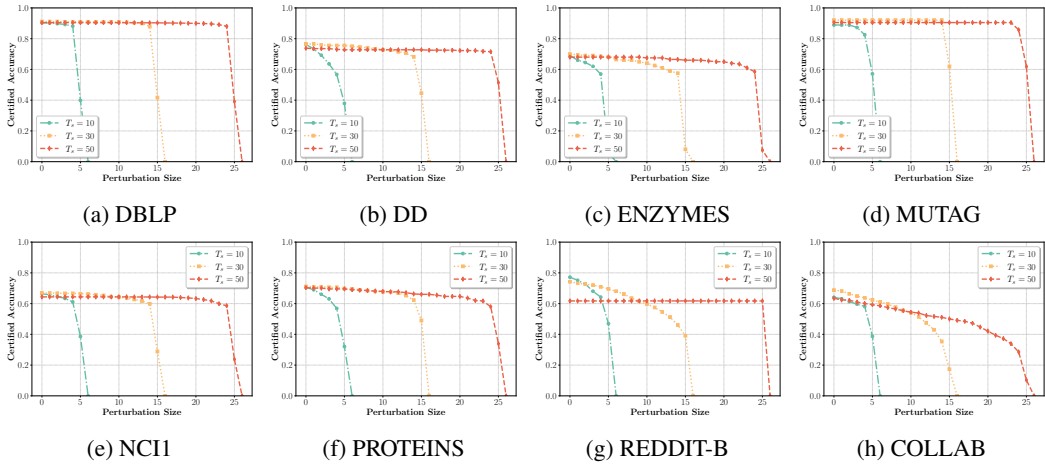

Figure 4: Impact of the number of groups $T_s$ on GNNCert for structure division.

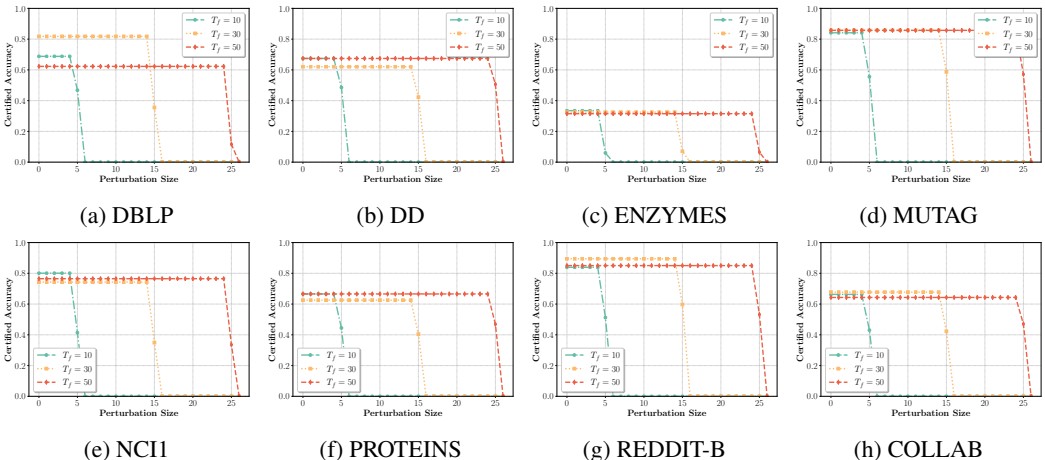

Figure 5: Impact of the number of groups $T_f$ on GNNCert for feature division.

**Our GNNCert is more efficient than existing defenses:** We also compare the efficiency of our GNNCert with existing defenses. Table 1 and 4 (in Appendix) compare the training and testing costs. We have two observations from the experimental results. First, our GNNCert is more efficient than existing methods in training the base graph classifier (note that we train the same number of epochs for the base classifier in our comparison). Our GNNCert is more efficient because, on average, sub-graphs used to train a base graph classifier for our GNNCert have less number of edges than the noisy graphs used by existing defenses. Second, we find that our GNNCert is more efficient in making predictions for testing graphs. The reason is that those three existing methods need to generate 1,000 noisy testing graphs for each testing graph while our method only needs to make predictions for 30 sub-graphs. In Figure 6 in Appendix D, we also set the number of noisy graphs to be the same as our method (i.e., use 30 noisy graphs). We find that their certified perturbation sizes for testing graphs significantly decrease.

**Training with sub-graphs improves the certified accuracy of GNNCert:** We train a base graph classifier using sub-graphs created from training graphs. Figure 3 compares the certified accuracy of our GNNCert when the base classifier is trained with and without sub-graphs. The result shows training with sub-graphs improves the certified robustness guarantees of our GNNCert. The reason is that the base graph classifier trained on sub-graphs is more likely to make correct predictions for sub-graphs created from a testing graph.

**Impact of $T_s/T_f$ on our GNNCert for structure/feature division:** Figure 4 and Figure 5 show the impact of $T_s$ and $T_f$ on the certified accuracy of our GNNCert with the structure division and feature

division, respectively. In general, we find that our GNNCert is more robust when $T_s$ (or $T_f$) increases. The reason is that our GNNCert could tolerate more perturbed sub-graphs as $T_s$ (or $T_f$) increases.

**Impact of hash function $\mathcal{H}$ on our GNNCert:** Figure 7 shows the impact of the hash function on our GNNCert. The experimental results show our GNNCert achieve similar certified accuracy for different hash functions, meaning our GNNCert is insensitive to the hash function.

**Impact of the architecture of graph neural network:** Figure 8 in Appendix shows the impact of the architecture of the base graph classifier on our GNNCert. The results show GNNCert is effective for different architectures, meaning our GNNCert is generally effective for base graph classifiers with different architectures.

**Our GNNCert can defend against structure and feature perturbations simultaneously:** Existing certified defenses for graph classification only focus on structure perturbation. By contrast, our GNNCert with structure-feature division could simultaneously resist both structure and feature perturbations. Figure 9, 10, 11, and 12 (in Appendix) show our experimental results under the default setting. Our experimental results demonstrate the effectiveness of our GNNCert for both graph structure and node feature perturbation.

We also conduct other experiments for ablation studies. Please refer to Appendix E for details.

## 5    DISCUSSION AND LIMITATION

**Other graph tasks:** We mainly focus on graph classification. We note that there are many other graph-relevant tasks such as node classification, link prediction, and community detection on graphs. Many existing studies (Chen et al., 2017; Zügner et al., 2018; Dai et al., 2018; Wang & Gong, 2019; Bojchevski & Günnemann, 2019a; Li et al., 2020; Sun et al., 2020; Ma et al., 2020) show those tasks are vulnerable to adversarial attacks as well. In response, many certified defenses (Bojchevski & Günnemann, 2019b; Jia et al., 2020; Schuchardt et al., 2020; Wang et al., 2021; Scholten et al., 2022) were proposed for these tasks. We leave the comprehensive study for those tasks as a future work.

**Node classification:** Our method could be extended to node classification. Given a node, we can view its ego network as a graph. We can use our GNNCert to build an ensemble graph classifier to predict the label of a node via its ego network. We use the white-box attack in Wan et al. (2021a) to craft a small perturbation such that a graph classifier makes incorrect predictions. For a standard graph classifier (GIN), the accuracy drops from 76% to 2% when adding/deleting at most 5 edges to each testing graph. Under the same perturbation, the accuracy of GNNCert is 68% for perturbed testing graphs. The results demonstrate that GNNCert is robust against perturbations for node classification.

**Node insertion/deletion:** Our GNNCert could be further extended to certify node insertion/deletion. In particular, we could divide nodes into different sub-graphs (using a hash function to calculate a group ID for each node), where each node only belongs to one group. Thus, inserting/deleting one node would influence at most one sub-graph, enabling us to derive the robustness guarantees. We added an experiment to validate this. On MUTAG dataset, our GNNCert could achieve a 59% certified accuracy when an attacker could arbitrarily add/delete one node (note that the attacker could arbitrarily add/delete edges for that node).

**Local structure:** For structure-feature division, our method could utilize local structure with a moderate number of sub-graphs. Each sub-graph may lose local structure information when the number of sub-graphs is large, i.e., there is a tradeoff between accuracy and robustness.

## 6    CONCLUSION

We propose GNNCert, a certified defense against graph structure and/or node feature perturbations on graph neural networks for graph classification. GNNCert provably predicts the same label for a testing graph once the graph structure and/or node feature perturbation is bounded. Our extensive experimental results on 8 benchmark datasets show GNNCert achieves better robustness guarantees and is more efficient than state-of-the-art defenses. It is an interesting future work to generalize GNNCert to certify the robustness for other tasks such as community detection.

ACKNOWLEDGMENTS

We thank the anonymous reviewers for their insightful reviews. This work was partially supported by the Cisco Research Award and National Science Foundation under grant Nos. ECCS-2216926, CCF-2331302, CNS-2241713 and CNS-2339686. Results presented in this paper were obtained using the Chameleon testbed supported by NSF. Any opinions, findings and conclusions or recommendations expressed in this material are those of the author(s) and do not necessarily reflect the views of the funding agencies.

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

**Table 2: Statistics of datasets.**

| Datasets | #Training graphs | #Testing graphs | #Classes |
|---|---|---|---|
| DBLP (Pan et al., 2013) | 12,971 | 6,485 | 2 |
| DD (Dobson & Doig, 2003) | 785 | 393 | 2 |
| ENZYMES (Hu et al., 2020) | 400 | 200 | 6 |
| MUTAG (Debnath et al., 1991) | 125 | 63 | 2 |
| NCI1 (Wale et al., 2008) | 2,740 | 1,370 | 2 |
| PROTEINS (Borgwardt et al., 2005) | 742 | 371 | 2 |
| REDDIT-B (Yanardag & Vishwanathan, 2015) | 1,333 | 667 | 2 |
| COLLAB (Yanardag & Vishwanathan, 2015) | 3,333 | 1,667 | 3 |

## A  PROOF OF THEOREM 1

Given a clean testing graph $G$, our GNNCert uses structure (or feature or structure-feature) division to divide it into $T$ sub-graphs (denoted by $\mathcal{G}_1, \mathcal{G}_2, \cdots, \mathcal{G}_N$). Given a base graph classifier $f$ and those $T$ sub-graphs, we use $N_c$ to denote the number of sub-graphs that are predicted as the label $c$ by the given base graph classifier. Formally, we have $N_c = \sum_{k=1}^{T} \mathbb{I}(f(G_k) = c)$, where $\mathbb{I}$ is the indicator function. We use $G^p$ to denote an adversarially perturbed graph. Moreover, we use $\mathcal{G}_1^p, \mathcal{G}_2^p, \cdots, \mathcal{G}_T^p$ to denote the $T$ sub-graphs created by our GNNCert for $G^p$ using structure (or feature or structure-feature) division. Recall that $M$ is the total number of sub-graphs in $\mathcal{G}_1^p, \mathcal{G}_2^p, \cdots, \mathcal{G}_T^p$ that are corrupted. Thus, at most $M$ sub-graphs among $\mathcal{G}_1^p, \mathcal{G}_2^p, \cdots, \mathcal{G}_T^p$ are different from those in $\mathcal{G}_1, \mathcal{G}_2, \cdots, \mathcal{G}_N$. Formally, we have $\sum_{k=1}^{T} \mathbb{I}(G_k \neq G_k^p) \leq M$. We use $N_c^p$ to denote the number of sub-graphs among $\mathcal{G}_1^p, \mathcal{G}_2^p, \cdots, \mathcal{G}_T^p$ that are predicted as the label $c$, i.e., $N_c^p = \sum_{k=1}^{T} \mathbb{I}(f(G_k^p) = c)$. Then, we have $N_c^p = \sum_{k=1}^{T} \mathbb{I}(f(G_k^p) = c) \leq \sum_{k=1}^{T} \mathbb{I}(f(G_k) = c) + M \leq N_c + M$. Similarly, we have $N_c^p \geq N_c - M$. Suppose $l$ is the predicted label of our GNNCert for the clean graph $G$. Then, our GNNCert predicts the label $l$ when we have the following: $N_l - M > \max_{c \in \{1,2,\cdots,C\} \setminus \{l\}} (N_c - \mathbb{I}(l < c) + M)$. In other words, we have $M < \frac{N_l - \max_{c \in \{1,2,\cdots,C\} \setminus \{l\}} (N_c - \mathbb{I}(l<c))}{2}$.

## B  PROOF OF THEOREM 2

We reuse the notations used in the proof of Theorem 1. Our key idea of the proof is to construct such a base graph classifier $f'$ such that the predicted label changes when $M^p + 1$ sub-graphs are corrupted. In particular, we select $M^p + 1$ sub-graphs among $\mathcal{G}_1, \mathcal{G}_2, \cdots, \mathcal{G}_N$ that are predicted as the label $l$ and corrupt them. Moreover, we let the base graph classifier $f'$ to predict the label $s = \arg \max_{c \neq l} (N_c - \mathbb{I}(l \leq c))$ for those corrupted sub-graphs. Then, we have $N_l^p = N_l - M^p - 1$ and $N_s^p = N_s + M^p + 1$. Since $M^p$ is the largest integer such that $N_l - M^p > \max_{c \in \{1,2,\cdots,C\} \setminus \{l\}} (N_c - \mathbb{I}(l < c) + M^p)$ holds. Thus, we have $N_l - (M^p + 1) \leq \max_{c \in \{1,2,\cdots,C\} \setminus \{l\}} (N_c - \mathbb{I}(l < c) + M^p + 1)$. In other words, the $l$ is not guaranteed to be predicted with our constructed base graph classifier $f'$.

## C  DETAILS OF DATASETS

Table 2 shows the statistics of the 8 datasets used in our evaluation.

## D  COMPARING OUR GNNCERT WITH THREE BASELINES WHEN SETTING THEIR NUMBER OF NOISY GRAPHS TO BE 30

We compare the computation cost and the certified accuracy of our GNNCert with the three compared methods when we generate 30 noisy graphs for the three compared methods. Table 3 compares the computation costs for each testing graph. We find that those defenses have similar computation costs.

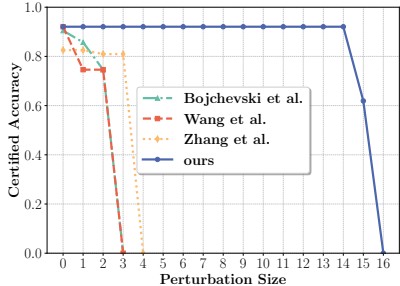

**Figure 6: Comparing the certified accuracy of our GNNCert when we generate 30 noisy graphs for the three compared methods. The dataset is MUTAG.**

However, the certified accuracy of those three methods drop very quickly as shown in Figure 6. The reason is that those defenses utilize Monte Carlo sampling to estimate the certified perturbation size. Thus, they need to generate many noisy graphs for each testing graph to accurately estimate the certified perturbation size.

# E    OTHER EXPERIMENT RESULTS

**Impact of the order of Node ID:** We note that each node has a node ID. We always put the node with a smaller ID first in our experiments. We also add an experiment to put the node with a larger ID first on MUTAG dataset. The certified accuracy is 92% (smaller ID first) and 92% (larger ID first) when an attacker arbitrarily adds/deletes one edge, respectively. The results demonstrate that our method is insensitive to the orders of nodes.

**Varying $N$ depending on the graph size:** We use a fixed number of sub-graphs on 8 datasets to be consistent. Our robustness guarantee against graph structure and node feature perturbation still holds when varying $N$ based on the number of nodes in a graph. For instance, we could set $T_s = 0.3 * \#$nodes in a graph On MUTAG dataset, the certified accuracies are 90% and 54% when an attacker could arbitrarily add/delete 1 and 5 edges.

**Comparing the accuracy of our ensemble graph classifier with standard graph classifier:** We conduct experiments to compare the accuracy of the normally trained GIN and GNNCert with GIN trained on sub-graphs. We have the following observations from the experimental results. On MUTAG, DD, COLLAB, DBLP, ENZYMES, PROTEINS datasets, the accuracy of GNNCert is at most 3% lower than that of normally trained GIN. On REDDIT and NCI1 datasets, the accuracy of GNNCert is at most 15% lower than that of normally trained GIN. The reason is that there is a tradeoff between accuracy without attacks and robustness. We note that such tradeoff widely exists for certified defenses (Cohen et al., 2019; Lecuyer et al., 2019; Levine & Feizi, 2020b). Our GNNCert (with GIN as the base classifier) reduces to GIN when $T_s = 1$. To reduce the accuracy drop, we could set a smaller $T_s$ in practice.

**Comparing the accuracy of base classifiers for different certification methods:** We also compare the accuracy of base classifiers for different certified methods. We find that, the accuracy of GNNCert's base classifier is higher than those of other defenses, which explains why our GNNCert outperforms them. For instance, on MUTAG dataset, the accuracy of GNNCert's base classifier is 91.85%. By contrast, the accuracy of base classifiers for Bojchevski et al., Wang et al., Zhang et al. are 67.88%, 88.89%, and 82.59%, respectively.

Table 3: **Comparing the computation cost (s) when we generate 30 noisy graphs for three compared methods. The dataset is MUTAG.**

| Compared method | Total (s) |
|---|---|
| Bojchevski et al. | 8.42 |
| Wang et al. | 12.62 |
| Zhang et al. | 12.80 |
| Ours | 7.94 |

Table 4: **Comparing the computation costs of GNNCert and existing defenses on multiple datasets.**

| Method | PROTEINS | | ENZYMES | | NCI1 | |
|---|---|---|---|---|---|---|
| | Train (s) | Test (s) | Train (s) | Test (s) | Train (s) | Test (s) |
| Bojchevski et al. (2020) | 2,351 | 421 | 1,930 | 157 | 1,811 | 920 |
| Wang et al. (2021) | 2,132 | 788 | 1,760 | 2,629 | 1,703 | 4,488 |
| Zhang et al. (2021b) | 1,575 | 196 | 1,427 | 95 | 1,357 | 525 |
| GNNCert | 829 | 13 | 760 | 11 | 815 | 22 |

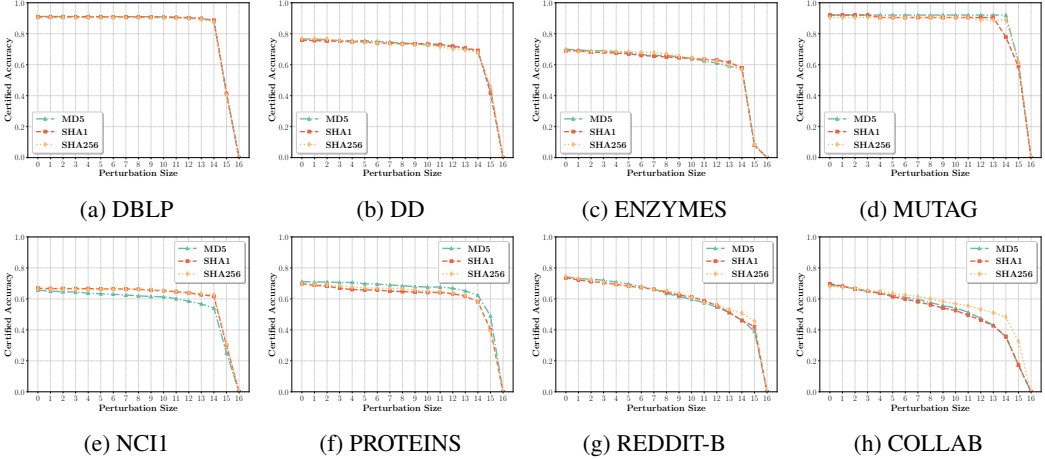

Figure 7: **Impact of the hash function $\mathcal{H}$ on GNNCert.**

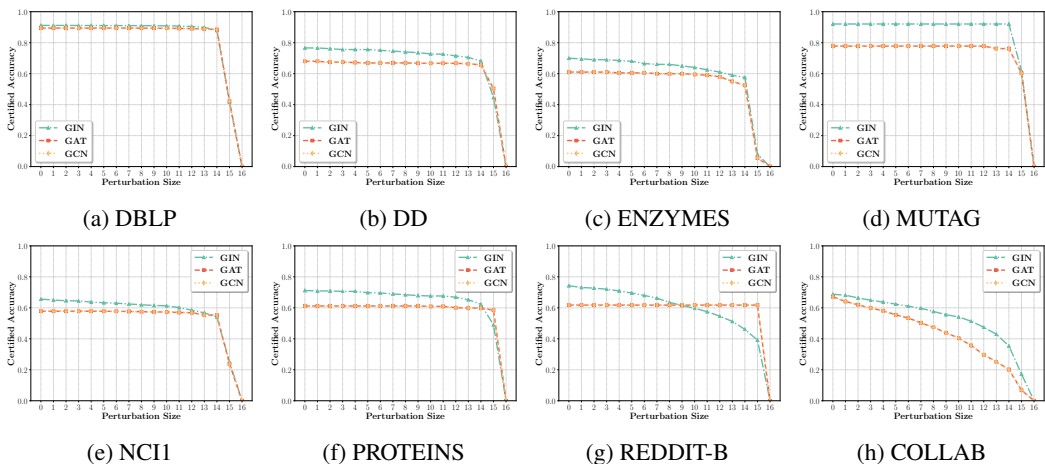

Figure 8: **Impact of the architecture of the base graph classifier on GNNCert.**

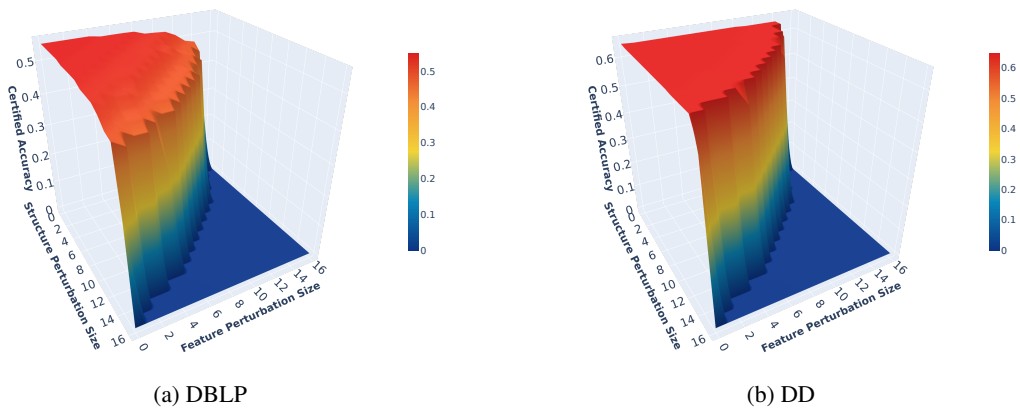

Figure 9: **Our GNNCert could resist structure and feature perturbation simultaneously.**

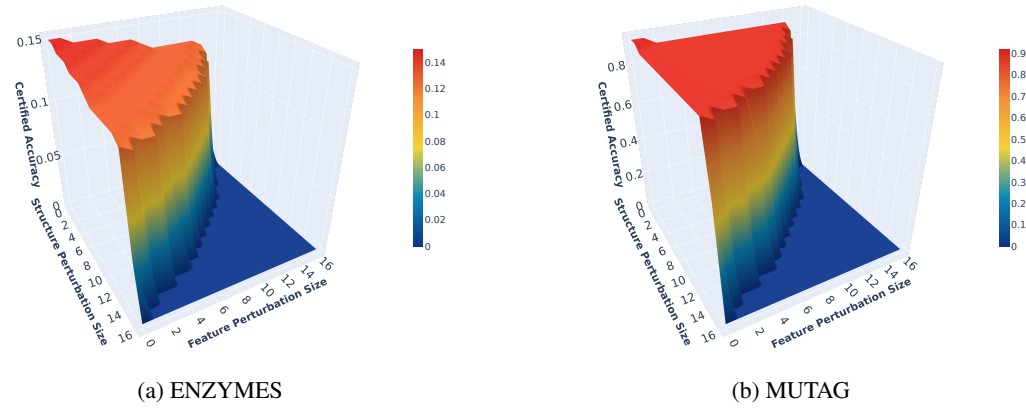

Figure 10: **Our GNNCert could resist structure and feature perturbation simultaneously.**

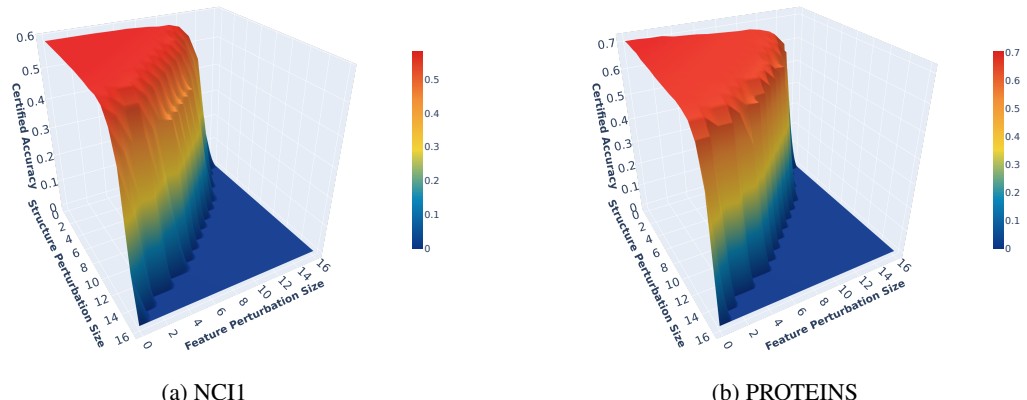

(a) NCI1

(b) PROTEINS

Figure 11: Our GNNCert could resist structure and feature perturbation simultaneously.

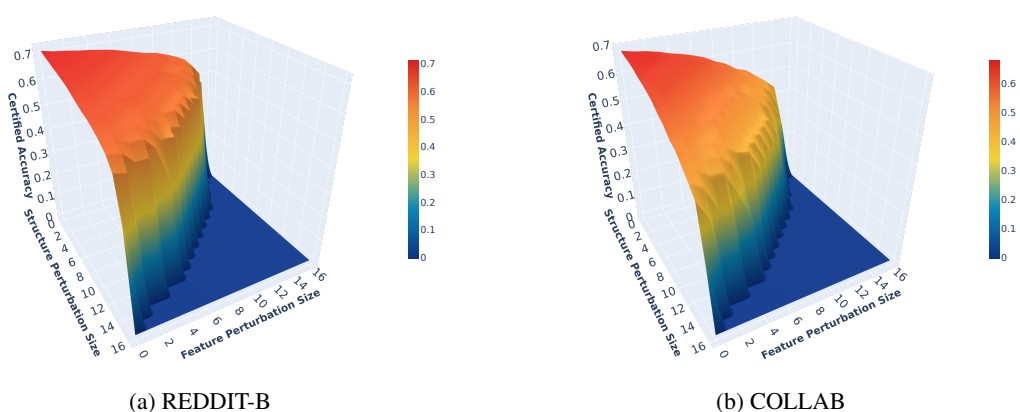

(a) REDDIT-B

(b) COLLAB

Figure 12: Our GNNCert could resist structure and feature perturbation simultaneously.

