# OpenReview forum: "GNNCert: Deterministic Certification of Graph Neural Networks against Adversarial Perturbations"
_ICLR.cc/2024/Conference — ICLR 2024 oral_

### Official Review · Reviewer_xWrk · 2023-10-28

**Soundness:** 3 good
**Presentation:** 1 poor
**Contribution:** 2 fair
**Rating:** 8
**Confidence:** 4

**Summary:**

This paper develops a graph classifier called GraphGuard, that is certifiably robust to structure perturbations and node feature perturbations. The classifier first partitions the graph edges and/or features using a hash function to yield $N$ sub-graphs. The sub-graphs are then classified by a base model (e.g., a graph neural net) to produce $N$ predictions, which are aggregated by majority vote to yield a classification for the entire graph. GraphGuard is shown to be robust to a specified number of edge perturbations and/or arbitrary node feature perturbations that depends on the number of sub-graphs and the margin between votes. Experiments on eight datasets demonstrate that GraphGuard achieves better certified accuracy that three baseline certified methods.

**Strengths:**

1. GraphGuard makes improvements in several dimensions compared to prior work on certified graph classification. Specifically, it yields deterministic guarantees, it is less computationally expensive, it can cover edge and node feature perturbations simultaneously, and it is shown to achieve superior certified accuracy. Given the breadth of these improvements, I think it will be of interest to the robustness/verification community.

1. The experiments are generally well-executed, apart from some issues mentioned below. It’s great to see results presented for several datasets and baselines. I appreciated the inclusion of experiments examining architectural choices, such as the architecture of the base neural network, and the choice of hashing function.

1. I liked the presentation of the paper overall. The writing was clear, leaving me with few doubts about the details. I found Figure 1 especially helpful in understanding the method.

**Weaknesses:**

1. The paper fails to cite prior work on derandomized smoothing which is strikingly similar to GraphGuard. For example, Levine & Feizi (2020) designed a certifiably robust classifier that also splits the input into sub-parts, classifies the sub-parts using a base model, and makes the final classification using majority vote. While their method is applied to image classification to certify against patch attacks, the fundamental design pattern (and analysis) is similar to GraphGuard. More recently, Hammoudeh & Lowd (2023) showed that the same design pattern can be used to achieve certified robustness against L0 perturbations (including patch attacks) at training-time and test-time. Given this context, GraphGuard could be viewed as an extension of derandomized smoothing to a new domain (graphs), which could impact the assessment of technical novelty. **Update 21/11: The authors have promised to address this issue. I have therefore raised my _Presentation_ score from 1 to 3.**

1. Although the experiments are comprehensive, I believe there is a crucial baseline missing: the base graph classifier (GIN) _without_ GraphGuard. Specifically, I would like to see a comparison of standard accuracy for GIN trained normally on full graphs, and GraphGuard with GIN trained on sub-graphs. This would allow for an assessment of GraphGuard’s impact on accuracy. This is important, as the standard accuracy of GraphGuard is fairly low for most datasets (around 70% based on Fig. 2)—it’s not clear whether this is due to GraphGuard or the inherent difficulty of the datasets. Incidentally, it may be interesting to report the accuracy of the base classifiers for all certified methods. Levine & Feizi (2020) found much higher base classifier accuracies for derandomized smoothing (which is similar to GraphGuard) than randomized ablation (which is similar to Zhang et al., 2021b). The same explanation may apply here.

1. While GraphGuard outperforms prior methods in terms of certified accuracy, the standard accuracy is rather low (at around 70% for 6 of the 8 datasets). This has a bearing on the significance of the paper in my view, as the sacrifice in terms of accuracy seems to high to be practical. It’s also worth noting that the variant of GraphGuard that protects against structure and feature perturbations simultaneously suffers a severe accuracy drop for some of the datasets (DBLP and ENZYMES). For these datasets, the classifiers are no better than random based on the standard accuracy in Figs. 9 and 10.

Minor:
1. The idea of training the base classifier on sub-graphs is claimed to be a contribution. However this is standard practice in randomized smoothing. It would be unusual not to train in this way.
1. Section 4.2: should alpha be 0.001?

**References:**

- Levine & Feizi, “(De)Randomized Smoothing for Certifiable Defense against Patch Attacks,” NeurIPS 2020. https://proceedings.neurips.cc/paper_files/paper/2020/file/47ce0875420b2dbacfc5535f94e68433-Paper.pdf

- Hammoudeh & Lowd, “Feature Partition Aggregation: A Fast Certified Defense Against a Union of $\ell_0$ Attacks,” AdvML-Frontiers 2023. https://openreview.net/forum?id=NX5Nxrz6PV

**Questions:**

1. The number of sub-graphs $N$ seems to be fixed for all inputs. Would it make sense to vary $N$ depending on the graph size (e.g., number of nodes)?
1. There doesn’t appear to be any restrictions placed on the hash function. I wonder whether it must be independent of the input’s edge structure and node features? Otherwise the certificate may not hold?
1. Is it possible to certify robustness against node insertion/deletion using this approach?
1. Does GraphGuard prevent the classifier from exploiting local structure, since each sub-graph tends to be sparse (both in terms of edges and node features)?

---

> ### Author Response · Authors · 2023-11-17
> **Response 1 to the Reviewer xWrk**
>
> We thank the reviewers for appreciating paper writing and the constructive comments
>
> **Question-1: Vary $N$ depending on the graph size.**
>
> Thanks for pointing this out. We use a fixed number of sub-graphs on 8 datasets to be consistent. Our robustness guarantee against graph structure and node feature perturbation still holds when varying $N$ based on the number of nodes in a graph. For instance, we could set $T_s=0.3 * \\#nodes\ in\ a\ graph$. On MUTAG dataset, the certified accuracies are 90\% and 54\% when an attacker could arbitrarily add/delete 1 and 5 edges.  We will add the discussion to our paper.
>
> **Question-2: No restrictions on the hash function (i.e., independent of edge structure and node features).**
>
> We don’t place restrictions on the hash function because we aim to resist arbitrary adversarial attacks (with bounded edge/node feature perturbations). If the hash function depends on the input’s edge structure and node features, then it could potentially be influenced by the attacker, making it more challenging to derive formal robustness guarantees against arbitrary attacks. We will add the discussion. Note that our GraphGuard is compatible with any hash function. Figure 6 shows our GraphGuard achieves similar performance with different hash functions.
>
> **Question-3: Certify robustness against node insertion/deletion.**
>
> Thanks for pointing it out. Our GraphGuard could be further extended to certify node insertion/deletion. In particular, we could divide nodes into different sub-graphs (using a hash function to calculate a group ID for each node), where each node only belongs to one group. Thus, inserting/deleting one node would influence at most one sub-graph, enabling us to derive the robustness guarantees. We added an experiment to validate this. On MUTAG dataset, our GraphGuard could achieve a 59\% certified accuracy when an attacker could arbitrarily add/delete one node (note that the attacker could arbitrarily add/delete edges for that node).
>
> **Question-4: Local structure.**
>
> We note that our GraphGuard with structure (or feature) division could utilize all node features (or entire graph structure). For structure-feature division, our method could utilize local structure with a moderate number of sub-graphs. We agree with the reviewer that each sub-graph may lose local structure information when the number of sub-graphs is large, i.e., there is a tradeoff between accuracy and robustness.

---

> > ### Author Response · Authors · 2023-11-17
> > **Response 2 to the Reviewer xWrk**
> >
> > **Comment-1: Prior work (Levine & Feizi (2020) and Hammoudeh & Lowd (2023)).**
> >
> > Thanks for pointing them out. We agree with the reviewer that splitting&predictions&majority vote is a general randomized smoothing-based framework for certification. The key challenge is how to tailor this framework for different domains. For instance, (de)randomized smoothing (Levine & Feizi (2020)) divides an image into different blocks or bands, which are customized for images. Hammoudeh & Lowd (2023) extend this general framework to certify $\ell_0$-norm perturbation for image and tabular data (all inputs have the same size) as mentioned by the reviewer.  By contrast, we focus on the graph domain, where the number of nodes and edges varies in each graph, making existing methods inapplicable. We develop three different methods to divide a graph into different sub-graphs. Our designs of three division methods (e.g., structure, feature, structure-feature division) are tailored to the graph domain. We will cite and discuss the difference with Levine & Feizi (2020) and Hammoudeh & Lowd (2023) as suggested.
> >
> > **Comment-2: Comparison of the accuracy of the normally trained GIN and GraphGuard with GIN trained on sub-graphs.**
> >
> > Thanks for the suggestions. We add experiments to compare the accuracy as suggested. We have the following observations from the experimental results. On MUTAG, DD, COLLAB, DBLP, ENZYMES, PROTEINS datasets, the accuracy of GraphGuard is at most 3\% lower than that of normally trained GIN. On REDDIT and NCI1 datasets, the accuracy of GraphGuard is at most 15\% lower than that of normally trained GIN. The reason is that there is a tradeoff between accuracy without attacks and robustness (such tradeoff widely exists for certified defenses). Our GraphGuard (with GIN as the base classifier) reduces to GIN when $T_s=1$. To reduce the accuracy drop, we could set a smaller $T_s$ in practice. We will add the results to our paper and discuss this tradeoff.
> >
> > We also compare the accuracy of base classifiers for different certified methods. As mentioned by the reviewer, the accuracy of GraphGuard’s base classifier is indeed higher than those of other defenses, which explains why our GraphGuard outperforms them. For instance, on MUTAG dataset, the accuracy of GraphGuard’s base classifier is 91.85\%. By contrast, the accuracy of base classifiers for Bojchevski et al., Wang et al., Zhang et al. are 67.88\%, 88.89\%, and 82.59\%, respectively. We will add the explanation. Moreover, we will discuss our observation is consistent with that in the image domain (Levine & Feizi (2020)).
> >
> > **Comment-3: The accuracy drop for protecting against structure and feature perturbations simultaneously on DBLP and ENZYMES.**
> >
> > We agree with the reviewer that the accuracy of our GraphGuard drops on some datasets such as DBLP and ENZYMES for structure-feature division. The reason is that we aim to certify structure and feature perturbations simultaneously. Additionally, we consider an attacker could arbitrarily manipulate the features of a number of nodes. In other words, we consider a very strong attacker. To reduce the accuracy drop, we could divide a graph into less number of sub-graphs. For instance, on DBLP dataset, our GraphGuard achieves a 76\% accuracy when dividing each graph into 9 sub-graphs ($T_s=3$, $T_f=3$).
> >
> > **Comment-4: Training the base classifier on sub-graphs.**
> >
> > We agree with the reviewer that this is standard practice in randomized smoothing. We will tone down our claim as suggested.
> >
> > **Comment-5: $\alpha$ should be 0.001.**
> >
> > We really appreciate the reviewer for pointing this out. We will fix the typo.

---

> > > ### Comment · Reviewer_xWrk · 2023-11-20
> > >
> > > Thank you for the comprehensive response.
> > >
> > > The primary weakness for me was the omission of prior work on derandomized smoothing. While I agree that there is a challenge in extending derandomized smoothing to other domains, I think it is essential to cite prior work that serves as a foundation. Given the authors have promised to cite and discuss Levine & Feizi (2020) and Hammoudeh & Lowd (2023), I will consider revising my score.

---

> > > > ### Author Response · Authors · 2023-11-20
> > > >
> > > > We really appreciate the reviewer for reading our response. We totally agree with the reviewer that prior work serves as a foundation for our work. We will definitely cite and discuss the prior work (Levine & Feizi (2020) and Hammoudeh & Lowd (2023)) in the next version as promised. We thank the reviewer for considering revising the score.

---

> > ### Comment · Reviewer_xWrk · 2023-11-20
> >
> > Regarding restrictions on the hash function: The following sentences in the response seem contradictory to me:
> >
> > > We don’t place restrictions on the hash function because we aim to resist arbitrary adversarial attacks
> >
> > > If the hash function depends on the input’s edge structure and node features, then it could potentially be influenced by the attacker, making it more challenging to derive formal robustness guarantees against arbitrary attacks.
> >
> > The 1st sentence says there are no restrictions, while the 2nd sentence says there are restrictions (in particular the hash function must not depend on the edge structure/node features). Based on my understanding, I believe the second sentence is correct. If so, the restriction should be clearly stated in the paper.

---

> > > ### Author Response · Authors · 2023-11-20
> > >
> > > We are so sorry for the confusion due to our misunderstanding. Yes, the understanding of the reviewer is correct: the hash function must not depend on the edge structure/node features. We will clearly discuss this point in the paper as suggested.

---

> > > > ### Comment · Reviewer_xWrk · 2023-11-21
> > > >
> > > > Thanks for clarifying. Given the authors have promised to address several weaknesses (missing prior work, assumptions on the hash function, and missing accuracy of GIN baseline and base classifier) I have decided to increase my overall score from 5 to 8.

---

> > > > > ### Author Response · Authors · 2023-11-21
> > > > >
> > > > > We really appreciate the reviewer for the constructive feedback, which significantly improves the quality of our paper.

---

### Official Review · Reviewer_nFKx · 2023-10-30

**Soundness:** 4 excellent
**Presentation:** 4 excellent
**Contribution:** 4 excellent
**Rating:** 8
**Confidence:** 4

**Summary:**

The paper introduces "GraphGuard," a certified defense mechanism for graph classification, designed to protect against adversarial perturbations in both graph structure and node features. Unlike existing defenses that lack robustness or have high computation costs, GraphGuard offers deterministic robustness guarantees. Through evaluations on 8 benchmark datasets, GraphGuard is shown to outperform current state-of-the-art methods.

**Strengths:**

**Originality**: The paper stands out for its innovative use of hashing to create subgraphs that distinguish between different types of perturbations, namely structure, feature, and both. The deterministic robustness guarantee presented is a fresh and valuable approach in the domain of certified defenses. Furthermore, the incorporation of both structure and node features sets the work apart, as few studies consider node features in this context.

**Quality**: The paper is grounded in solid theory and its effectiveness is corroborated by empirical evaluations.

**Clarity**: The paper is articulate and straightforward. The authors proactively address and clarify potential ambiguities in each section, ensuring a smooth reading experience.

**Significance**: Given the growing prominence of graph neural networks across various applications, addressing security in downstream graph tasks is paramount. This study zeroes in on the security of the graph classification task, potentially paving the way for enhanced security measures in other tasks like node classification and link prediction, emphasizing a deterministic robustness guarantee.

**Weaknesses:**

* On Page 4, at the end of line 4: Shouldn't the notation be $H(.)$%$T_s + 1$?

* Based on my understnading $(S_{v_t} \bigoplus S_{v_j}) \neq (S_{v_j} \bigoplus S_{v_t})$. If that is correct, then it seems in structure division explained in 3.3.1 the same edge can have two different hash values depending on which end node is visited first. Can the authors clarify this?

* The hashing process, while detailed, lacks clarity on one aspect: How are node features transformed into string representations for the hash function? Given the diverse nature of node features in graphs, understanding this process, especially for features requiring unique treatments, is essential.

* Could the authors provide an intuitive explanation for the preference of hash-based subsampling over methods like $\tau$ fraction-based sampling (referenced in the benchmarks) or other similar techniques? What advantages does this approach offer and what are its potential limitations?

* How do the authors determine the values for $T_s$ and $T_f$? It would be beneficial if this determination were associated with specific graph properties (e.g., the number of edges, nodes, diameter, path length, clustering coefficient, etc.), as this would guide potential users of this defense strategy in tailoring it to their datasets.

**Questions:**

See above.

---

> ### Author Response · Authors · 2023-11-17
> **Response to the Reviewer nFKx**
>
> We thank the reviewers for appreciating the novelty, significance, solid theory, and constructive comments!
>
>
> **Comment-1: The notation at the end of the line 4 on Page 4.**
>
> Thanks for pointing it out. It is indeed $H(\cdot)\\%T_s+1$. We will fix the typo.
>
>
> **Comment-2: The hash value depends on which node is visited first.**
>
> Sorry for the confusion. We note that each node has a node ID. We always put the node with a smaller ID first. We also add an experiment to put the node with a larger ID first on MUTAG dataset. The certified accuracy is 92\% (smaller ID first) and 92\% (larger ID first) when an attacker arbitrarily adds/deletes one edge, respectively. The results demonstrate that our method is insensitive to the orders of nodes. We will clarify and add results to our paper.
>
>
> **Comment-3: How node features are transformed into string representations.**
>
> Node features could be represented as a feature vector, where each feature value is a real number. We transform each feature value into a string. For instance, if the feature value is 0.12, we can transform it into the string “0.12”. Then, we concatenate the strings for different feature values. We will clarify as suggested.
>
>
> **Comment-4: Preference of hash-based sub-sampling over $\tau$ fraction-based sampling.**
>
> Our hash-based sub-sampling has the following advantages. First, our hash-based sub-sampling enables our defense to certify feature perturbation as well as both feature and structure perturbations for graph classification. By contrast, existing sub-sampling methods cannot. Second, hash-based sub-sampling divides a graph into sub-graphs deterministically while $\tau$ fraction-based sampling generates sub-graphs probabilistically. As a result, our method could provide deterministic robustness guarantees while existing methods cannot due to the randomness in their sub-sampling. Third, our hash-based division makes our defense very efficient (as shown in Table 1). The reason is that our hash-based sub-sampling divides a graph into a moderate number of (e.g., 30) sub-graphs. By contrast, existing  $\tau$ fraction-based sampling requires sampling hundreds of sub-sampled graphs to estimate the robustness guarantees (as they utilize Monte-Carlo to estimate them). Our experimental results in Figure 8 (in Appendix) show the certified accuracy of existing methods is much worse when reducing the number of sub-sampled graphs for them.
>
> In general, there is a tradeoff between utility (measured by the accuracy without attacks) and robustness. One limitation of our defense is that it also has such a tradeoff. For instance, when the number of groups is very large, the accuracy without attacks of our defense slightly drops. The experimental results show our defense achieves a better tradeoff than existing methods.
>
>
> **Comment-5: How to determine the values of $T_s$ and $T_f$.**
>
> In our experiment, we use a fixed $T_s=30$ and $T_f=30$ for all 8 datasets to be consistent. Additionally, we show the impact of $T_s$ and $T_f$ in Figures 4 and 5. Our results show GraphGuard could achieve better robustness guarantees when $T_s$ and $T_f$ are larger. We could set $T_s$ and $T_f$ based on the number of edges and nodes, respectively. In particular, we could set a larger $T_s$ and $T_f$ when the number of edges and nodes in a graph is larger to ensure better robustness.

---

> > ### Comment · Reviewer_nFKx · 2023-11-22
> >
> > Thanks for answering most of my questions. The only question that I still am not certain about is how to connect the hyperparameters $T_s$ and $T_f$ to some network property such that it becomes easier for someone to adapt the method to their own data (instead of iteratively going through different sets that might work for them). However, this is a minor issue and does not change my overall rating of the paper. I still see this as a good fit to the conference.

---

> > > ### Author Response · Authors · 2023-11-23
> > >
> > > We really appreciate the reviewer for the comment. We will add the discussion on setting hyperparameters $T_s$ and $T_f$. Additionally, we will conduct more experiments to connect $T_s$ and $T_f$ with graph properties  (e.g., the number of edges and nodes as mentioned by the reviewer) to further optimize our results. We thank the reviewer again for the constructive feedback, which significantly improves the quality of our work.

---

### Official Review · Reviewer_LqWt · 2023-11-01

**Soundness:** 3 good
**Presentation:** 4 excellent
**Contribution:** 3 good
**Rating:** 8
**Confidence:** 3

**Summary:**

This paper claims to provide a robustness of graph classifications under adversarial attacks. It proposes a hashing method for partitioning the graph into several (overlapping) subgraphs and then ensembling them into a final classifier. The paper derives theoretical and empirical bounds on the robustness of the classifier, demonstrating that it behaves better than the state-of-the-art. No additional computational overhead is added on the classifier compared to state of the art.

**Strengths:**

The theory of the paper is solid, and the experiments prove the point that the authors want to make.
The presentation is pretty clear and easy for the reader to follow.

**Weaknesses:**

As an expert in the security domain with a strong background in signal processing, the paper looks more like building a classifier robust to noise rather than a defense method against adversarial attacks. When it comes to internet security applications, in order to add value, someone has to work on real case scenarios where the deception in building the graph can be realistic. Here, the benchmarks are very weak in terms of real-case scenarios. In cases like malware or DNS graphs, etc, the deception models have to be very sophisticated and realistic. I understand that this is more like an ML theory paper, but it really has no value in real life.
Another problem that I have with this paper is the use of the term "adversarial attack." A malicious agent tries to minimize the perturbation so that the perturbed graph is as close as possible to the initial one. Either through a black or white box attack, the agent tries to reverse engineer the classifier so that a small perturbation can lead to a big change in the output; they might choose to change, let's say, one feature or connection that will have a big impact. I don't see that study here. Instead, the authors claim that we will make it robust to a certain type of noise.

**Questions:**

My single question that will affect my decision is the following?
Was the perturbation uniform noise?
Can the authors pick at least one or two datasets and experiment with types of noise that would make case in a realistic malicious agent scenario?

---

> ### Author Response · Authors · 2023-11-17
> **Response to the Reviewer LqWt**
>
> We thank the reviewer for appreciating the solid theoretical contributions and constructive comments!
>
> **Question-1: Was the perturbation uniform noise?**
>
> Sorry for the confusion. The graph classifier built by our defense could resist arbitrary perturbations (e.g., uniform noise perturbation, carefully crafted perturbation by a malicious agent using a black or white-box attack) to a graph, once the total number of perturbed edges or number of nodes with perturbed features are bounded. For instance, with structure division, we derive a lower bound of the accuracy (i.e., certified accuracy) that our graph classifier could achieve on a testing dataset when an attacker could arbitrarily add/delete at most a number of edges (called perturbation size) to each testing graph. Take Figure 2 as an example. Our results show the accuracy drop of our graph classifier is within 3\% when an attacker arbitrarily adds/deletes one connection to each testing graph. We will clarify.
>
> **Question-2: Add datasets for a realistic malicious agent scenario.**
>
> We test a real-world Twitter dataset [1] as suggested, where the task is to classify whether a user is fake or benign based on its ego network. We use the white-box attack in [2] to craft a small perturbation such that a graph classifier makes incorrect predictions. For a standard graph classifier (GIN), the accuracy drops from 76\% to 2\% when adding/deleting at most 5 edges to each testing graph. Under the same perturbation, the accuracy of our GraphGuard is 68\% for perturbed testing graphs. The results demonstrate that our GraphGuard is robust against carefully crafted perturbations. We will incorporate the results and corresponding discussion into our paper.
>
> [1] Zhang et al. “Backdoor Attacks to Graph Neural Networks”, 2022.
>
> [2] Wan et al. "Adversarial attacks on graph classification via bayesian optimisation", 2021.

---

> > ### Comment · Reviewer_LqWt · 2023-11-23
> > **Updating the score**
> >
> > Thanks for the clarifications, I will raise my score

---

> > > ### Author Response · Authors · 2023-11-23
> > >
> > > We really appreciate the reviewer for reading our response and providing constructive feedback on our work!

---

### Meta-Review · Area_Chair_zV9D · 2023-12-10

**Metareview:**

The reviewers found the paper to be overall well-executed, having a solid theoretical component and complemented by convincing, well-designed experiments.

On the weaknesses: reviewers raised concerns that required clarifications, and the authors sufficiently addressed them in their rebuttal. The authors are strongly encouraged to incorporate their reponses to the final version. The most important are to include and contrast to Levine & Feizi (2020)  and  Hammoudeh & Lowd (2023) in their related work, stating explicitly any assumptions missing regarding the hashing function used, and including the base classifier (GIN) without GraphGuard (experiments with which were reported in the rebuttal).

**Justification For Why Not Higher Score:**

N/A

**Justification For Why Not Lower Score:**

The paper received 3 Accepts (8). One reviewer thought that this could be a spotlight, this could be moved down, but another reviewer supported this as an oral.

---

### Decision · Program_Chairs · 2024-01-16

Accept (oral)